# Post-Operative Patients’ Satisfaction and Quality of Life Assessment in Adult Patients with Obstructive Sleep Apnea Syndrome (OSAS)

**DOI:** 10.3390/ijerph19106273

**Published:** 2022-05-21

**Authors:** Diego Sergio Rossi, Funda Goker, Federico Cullati, Alessandro Baj, Daniele Pignatelli, Aldo Bruno Gianni, Massimo Del Fabbro

**Affiliations:** 1Dental and Maxillo-Facial Surgery Unit, IRCCS Ca Granda Ospedale Maggiore Policlinico di Milano, Via Francesco Sforza 35, 20122 Milan, Italy; diego.rossi@policlinico.mi.it (D.S.R.); ricofede22@gmail.com (F.C.); alessandro.baj@unimi.it (A.B.); aldo.gianni@unimi.it (A.B.G.); 2Department of Biomedical, Surgical and Dental Sciences, University of Milan, 20122 Milan, Italy; massimo.delfabbro@unimi.it; 3School of Medicine, University of Milan, 20122 Milan, Italy; danielepignatelli1997@gmail.com; 4IRCCS Orthopedic Institute Galeazzi, Via Riccardo Galeazzi 4, 20161 Milan, Italy

**Keywords:** OSAS, orthognathic surgery, maxillomandibular advancement surgery, bi-maxillary surgery

## Abstract

Background: The treatment for severe OSAS includes maxillomandibular advancement surgical option in selected cases. The aim of this study was to evaluate the post-operative impact of bimaxillary surgery on satisfaction and consequently the quality of life of these patients. Methods: This study included 18 patients with severe OSAS who received maxillomandibular advancement surgery. Patients were divided into Group A (operated by CAD/CAM) and Group B (conventional surgery). The impact of bimaxillary surgery on satisfaction and quality of life of these patients was evaluated by utilizing post-operative life quality and Rustemeyer’s patient-satisfaction-based survey. Results: A total of 18 adult OSAS patients (Group A: 11 patients, Group B: 7 patients) with a mean age of 44.39 years (SD ± 9.43) were included. Mean follow-up period was 32.64 months (SD ± 21.91). No intra-operative complications were seen in any patients. Post-operative complication was seen in one patient and the mandible did not integrate. According to the results, overall post-operative satisfaction score was 79.72% (SD ± 9.96). There was no significant difference among those in Group A and Group B. Conclusions: Maxillomandibular advancement surgery seems to be beneficial in terms of patients’ satisfaction in severe adult OSAS patients and can be considered as a valuable option in selected cases.

## 1. Introduction

Obstructive sleep apnea syndrome (OSAS) is a universally frequent public health issue, which is characterized by episodes of partial or complete collapse of the pharynx during sleep [1,2,3,4]. In OSAS patients’ obstruction of the upper airway results with interrupting (apnea) or reducing (hypopnea) the flow of air, followed by transient awakening, that leads to the restoration of upper airway permeability. Clinical manifestations include headache, day-time sleepiness, concentration difficulties, and a decrease in cognitive performance [1]. Nocturnal symptoms include recurrent arousal during sleep, snoring, witnessed/perceived apneas. Moreover, these are commonly associated with negative health consequences and medical comorbid conditions [5]. According to the scientific literature, the prevalence of cardiovascular disease, hyperlipidemia, hypertension, respiratory disease, relative insulin resistance, cerebrovascular disease, and type 2 diabetes mellitus (DM), gastroesophageal reflux disease (GERD), obesity, depression and other psychiatric disorders, cognitive dysfunction, and migraine headaches are higher in OSAS patients [6,7,8,9,10,11,12,13,14,15,16,17,18,19,20]. Consequently, these patients suffer from a worsening of life quality [1,21,22].

This syndrome can be seen at any age group and a correct diagnosis of OSAS mostly depends on proper anamnesis [3]. There are also several diagnostic tests for the evaluation of sleep and accurate diagnosis. Among these tests, in-laboratory polysomnography (PSG) is considered as the gold standard diagnostic test for OSAS [2]. PSG uses multi-channel continuous recordings for evaluation such as electrocardiography, electromyography, electroencephalography, electro-oculography, respiration, nasal airflow, oximetry, snoring, the distribution of the stages of sleep, the number of awakenings, and the apnea/hypopnea index (AHI). AHI, which is defined as the average number of apneas and hypopneas per sleep hour, is being used to evaluate OSAS severity. An AHI score of <15 per hour indicates mild apnea, whereas AHI 15–30 per hour defines moderate apnea. AHI score which is greater than 30 shows very serious OSAS and in such cases, surgery is considered as a treatment option [2,3,23]. In patients with mild and moderate OSAS, oral appliance therapy such as mandibular advancement devices (MAD) and continuous positive airway pressure (CPAP) therapy are currently the most used treatment options [21,24,25,26,27,28,29,30].

The surgical options for severe OSAS treatment mentioned in literature include nasal surgery, uvulo-palato-pharyngoplasty, genioglossus advancement, and maxillomandibular advancement (MMA) [22,31,32]. Maxillomandibular advancement (MMA) surgery is an invasive procedure and the peri- and postoperative risks in this type of surgery include infection, inflammation, pain, malocclusion, and unsatisfactory cosmetic results. However, currently there is an increasing number of reports in literature about favorable and successful results with significant decreases in AHI values [33,34]. Today, MMA surgery is considered as an effective treatment for OSAS, in which an enlargement of the upper airway is achieved by physically expanding the facial skeletal framework [35,36,37,38,39].

Untreated OSAS can result in undesirable physiological, behavioral, and cognitive sequelae. Research reports highlight the fact that the patients with OSAS have poorer overall quality of life (QoL) when compared with healthy individuals [1,21,22,30]. Currently, there is increasing interest from the scientific community for post-operative QoL evaluation of OSAS patients [30,40,41,42,43]. Among the options, the OSA-18 questionnaire by Franco et al. [42] is the most widely used QoL survey aimed at paediatric OSAS patients and has been validated as an informative instrument. For adult patients, generic instruments, such as the Medical Outcome Survey, Short Form (SF)-36, are mostly being used along with some reports on disease-specific questionnaires [30,40,41,42,43,44,45,46,47,48]. However, there is still a limited number of reports in literature that investigate the post-operative QoL evaluations in such patients.

Rustemeyer et al. proposed a questionnaire about patients’ satisfaction and changes in quality of life after orthognathic surgery. This short form of survey was proposed as a post-operative tool for the evaluation of patients’ overall satisfaction, the relatives’, and friends’ opinions about the results of surgery, and aesthetic and masticatory improvements compared to before surgery [49,50].

This study aimed to explore relations between MMA surgery and QoL improvements in adult patients with OSAS. We hypothesized adults that had MMA surgery for OSAS treatment would experience improvements in their quality of life. For this purpose, a retrospective clinical study was performed on OSAS cases for the evaluation of MMA post-operative changes in quality of life and post-operative patient satisfaction.

## 2. Materials and Methods

This retrospective clinical study included 18 OSAS patients with severe OSAS (apnea hypopnea index AHI > 30 per hour). The aim was to evaluate the impact of maxillomandibular advancement surgery on satisfaction and quality of life (QoL) of these patients. For this purpose, Rustemeyer’s questionnaire (Table 1) was used to evaluate the overall satisfaction of the participants, the opinions of the relatives and friends about the results of the surgery, and aesthetic and masticatory improvements compared to before the surgery. Additionally, the results of post-operative quality-of-life questions specific for OSAS (Table 2) were assessed.

This study included 18 (1 female, 17 male) severe OSAS patients who underwent Le Fort I maxillary osteotomy and bilateral sagittal mandibular osteotomy between April 2016 and December 2021 at the Department of Oral and Maxillofacial Science, University of Milan. The study protocol was approved by the Ethics Committee of Fondazione IRCCS Ca’ Granda Ospedale Maggiore Policlinico, Regione Lombardia with date 09/03/2016 Ethics Committee of Milano Area B Act 1300/2015, Determinazione no: 421. This study followed the principles laid down in the Declaration of Helsinki on medical protocol and ethics.

38 severe OSAS patients who underwent MMA surgery at Ospedale Maggiore Policlinico were contacted by phone and were asked if they were willing to participate in the study. 18 severe OSAS patients (out of 38 consecutive patients) agreed to cooperate, and a Rustemeyer survey was obtained from these subjects to be evaluated.

Inclusion criteria: Patients with severe OSAS (apnea hypopnea index AHI > 30/h) who had MMA surgery at Ospedale Maggiore Policlinico and agreed to participate the study post-operatively by answering the Rustemeyer and quality of life surveys.

Exclusion criteria: Patients with mild OSAS (AHI 5–14/h) or moderate OSAS (AHI 15–30/h), and only retropalatal collapse cases. No other exclusion criteria were set.

### 2.1. Pre-Operative Preparation

Presurgical protocol included taking detailed health anamnesis from each patient with clinical and radiological examinations. Figure 1, Figure 2, Figure 3, Figure 4, Figure 5, Figure 6, Figure 7, Figure 8, Figure 9 and Figure 10 show representative pre-planning of one of the OSAS patients that received MMA surgical treatment.

### 2.2. Surgical Procedures

Pre-operatively, drug-induced sleep endoscopy (DISE) was performed in the operating room with an anesthesiologist, attended by an otorhinolaryngologist and a maxillofacial surgeon, to clearly identify the site of obstruction.

Under general anesthesia with nasotracheal intubation and local anesthesia with vasoconstrictor (4% articaine with 1:100,000 adrenalin), MMA surgery was performed. CAD/CAM (Computer-aided design and computer-aided manufacturing) patients were operated using “mandible-first approach”. In traditional MMA surgery, maxillary operations were done first.

#### 2.2.1. Mandibular Operation

Bilateral sagittal-splint osteotomies of the mandibular bone were performed with the aid of cutting guides using a piezoelectric or conventional saw instrument. Subsequently, pre-planned mandibular advancement was achieved and maintained with plates and osteosynthesis screws (either patient-specific CAD-CAM custom-made plates or plates).

#### 2.2.2. Maxillary Operation

Maxillary Le Fort I osteotomy was performed with the excision of any overlapping bone, as determined by pre-planning. The maxilla was moved to its new position (after anticlockwise rotation and advancement), which was established accurately by an occlusal splint attached to the mandible. Finally, the maxilla was fixed on each side with two L-shaped miniplates and bi-cortical screws.

The occlusion was maintained by an occlusal splint and elastic maxillomandibular fixation. Patients were admitted overnight into intermediate care and were followed in the general ward for 3 to 5 days.

### 2.3. Post-Operative Protocol

The standard follow-up regimen included routine weekly visits in the first month, then every 2 weeks in the second and third months, then monthly until the end of the first year. Patients wore the elastic maxillomandibular fixation apparatus for 24 h/day for 2 weeks, then overnight only for two weeks. They then underwent removal of the elastics and were allowed fluids and a soft diet for the following two weeks.

Potential complications, including pain, oedema, infection, nerve injury and paresthesia, problems with surgical wound healing and bony union, occlusal problems, and tooth loss, were addressed when present.

#### The Antibiotic and Medications Regimen

Augmentin 1 g (3 × 1, for 5 days), Ketoloprac (15 gtt 3 × 1, for 2 days) or Azithromycin 500 mg for 3 days in cases of allergy to penicillin. Paracetamol (3 × 1, for 2 days and continue in case of pain and fever), Pantoprozol (20 mg 1 × 1, for 5 days), Rinostill plus (or any other Aerosol with acetylesystein 3 × 1, for 4 days), Clorhexidine rinses (after meals).

### 2.4. Data Collection and Outcome Evaluation

Data collection included demographics, medical history, the Rustemeyer Questionnaire, and QoL forms. The primary outcome variables of this study were based on the survey and Questionnaire forms obtained. Additionally, results of the CAD/CAM and traditional surgery were compared.

A short six-item form of the Rustemeyer’s questionnaire [49,50] was used to assess the overall satisfaction of the participants, and the opinions of relatives and friends about the results of surgery, and aesthetic and masticatory improvements compared to before surgery (Table 1). An Italian version of the Rustemeyer’s questionnaire was not available, so it was translated into Italian.

The included patients did not have any pre-operative Quality of Life (QoL) Questionnaire. However, to make a post-operative comparison, answers to questions specific for OSAS (based on quality-of-life domains of the OSA-18 questionnaire) were obtained from each patient [42]. Further details about post-operative questions for QoL data collection for evaluation can be seen in Table 2.

### 2.5. Statistical Analysis

Statistical analysis was performed using GraphPad Prism 5.03 (GraphPad Software, Inc., La Jolla, CA, USA). Descriptive statistics of the data were done using mean values and standard deviation (SD) for quantitative variables normally distributed. Normality of distribution was evaluated through the d’Agostino and Pearson omnibus test. The comparison between traditional and CAD-CAM group for scores of each of the Rustemeyer questions was made using the non-parametric Mann–Whitney test for independent samples. The comparison for questions specific for OSAS (quality-of-life domains of OSA-18 questionnaire) was made using the Fisher’s exact test, given the low sample size. Comparison of BMI between groups pre- and post-surgery was made with unpaired Student’s *t*-test, and comparison between pre-and post-surgery was made with Student’s paired *t*-test. A probability value *p* = 0.05 was considered as the significance threshold.

## 3. Results

### 3.1. Study Groups

The study group consisted of 18 adult OSAS patients with a mean age of 44.39 ± 9.43 (standard deviation, SD) ranging from 24 to 59 years. Mean follow-up period after operation was 32.64 ± 21.91 months. The patients were divided into two groups as Group A: patients operated with CAD/CAM surgery (11 patients), and Group B: patients operated by traditional methods (7 patients). The demographics of the included patients are listed in Table 3. Pre-operative and post-operative BMI index were compared for each patient. As a result, there was a significant difference between pre-operative and post-operative BMI index (*p* = 0.042). However, there was no significant difference between groups.

#### 3.1.1. Complications

Post-operative complications were seen in six patients. In one patient a major problem occurred, and the mandible did not integrate after surgery. In this case, as a treatment, revision surgery was scheduled and performed (40 days after the first surgery), in which the osteosynthesis plaques were removed and replaced by new ones bilaterally. This patient experienced no other post-operative complications. Further details about surgical interventions and information about post-operative complications for each patient can be found in Table 4. No intra-operative complications were seen in any patients.

#### 3.1.2. Results of Rustemeyer’s Questionnaire

Overall patient post-operative satisfaction score averaged 79.72 ± 9.96% (post-operative satisfaction score in Group A: 81.5 ± 11%, and in Group B: 76.9 ± 8.9%). Patient satisfaction was not significantly different in CAD/CAM patients when compared to traditional surgery (*p =* 0.32). In Table 5, Rustemeyer’s questionnaire results are listed.

#### 3.1.3. Results of Post-Operative QoL Questionnaire

According to the results of comparison for questions specific for OSAS (quality of life domains of OSA-18 questionnaire), overall QoL results indicate an improvement following surgery. However, there was no significant difference between the two groups. There was a slightly significant (*p =* 0.04) reduction of the BMI in the post-op period (from 29.06 ± 4.53 to 27.65 ± 3.45). Further details on QoL questionnaire results can be found in Table 6.

## 4. Discussion

The OSAS Syndrome is a disease that has critical negative impacts on people’s lives. Currently, the frequency of OSAS has increased worldwide, and it is about 2–3 times more frequent in males than females. OSAS has multifactorial etiology, and the diagnosis of OSAS, which has a great impact on successful treatment results, is often neglected. Morphological features of the patients such as obesity, anatomical aspects of the jaws and airways, and posture during sleep are important predisposing factors [4]. There is a variety of treatment options mentioned in literature, mostly depending on the severity of the disease. In cases of severe OSAS and for patients who are not suitable for conservative OSAS therapies such as CPAP, surgical treatment is considered as an option [22,23,32]. The presence of untreated OSAS is associated with a poorer quality of life and is a critical risk factor for the development of various clinical diseases and mental disorders [4].

QoL questionnaires are increasingly recognized as an important health outcome measure in clinical medicine [40]. Over the last decades many QoL questionnaires have been proposed and used for evaluation of the impact of OSAS symptoms and to assess the post-operative improvements. Today, there are various QoL questionnaires available to evaluate the QoL improvements and to compare the pre-treatment situation and post-treatment outcomes. Among the options, the OSAS-specific “OSA-18 questionnaire” is widely accepted and validated as an informative instrument [30,41]. Health generic instruments such as SF 36 are also being widely used by researchers although there is still a limited number of reports evaluating QoL and patient satisfaction [30,40,41,42,43,44,45,46,47,48]. All these mentioned surveys are used to evaluate and compare the pre-/post-operative condition. However, for evaluating post-operative results in OSAS patients that did not fill out pre-operative surveys, there are no papers for evaluating patient satisfaction. In the opinion of the authors of this work, any data would be of importance for taking the surgical decision in adult OSAS patients, since the reports are quite limited.

The patients that participated in this work have not participated in any pre-operative quality of life questionnaire assessment. However, according to the opinion of the authors, the post-operative evaluation of these patients would be important to understand the impact of MMA surgery. For this purpose, to compare post-operative QoL, answers to questions specific for OSAS (quality-of-life domains of the OSA-18 questionnaire) were obtained from each patient to be evaluated.

Despite the technological and equipment progress that has made the orthognathic surgeries much faster and simpler than they used to be, patients’ dissatisfaction with the outcomes is still a common issue [51]. MMA surgery is considered as a highly aggressive invasive surgery, and risks and benefits should be assessed with caution before taking a decision in adult patients, especially in subjects with compromised health conditions. Limitations of this study include the limited number of the sample group, and no evaluation with cephalometric changes in hard soft tissue variables, the apnea/hypopnea index (AHI) changes, and no comparison between pre-/post-operative patient satisfaction. However, this paper might be valuable and helpful for clinicians making a critical decision for an adult OSAS patient with compromised health to evaluate the future risks and benefits of MMA surgery.

According to the Rustemeyer questionnaire results, overall patient satisfaction after surgery can be considered high, as 79.72% of the participants declared high positive results in terms of post-operative satisfaction. As can be seen in Table 5, the patients gave scores from 5 to 10 reflecting their satisfaction. Additionally, all the answers given to the Rustemeyer questionnaire were 5 or higher than 5, which cannot be considered as a total dissatisfaction for the residual 20.28% of the study group. Besides, patient satisfaction was statistically higher in CAD/CAM patients in terms of facial esthetics when compared to traditional surgery (*p* = 0.003).

The CAD/CAM surgical approach in orthognathic surgery represents several advantages when compared with conventional surgical planning, such as the visualization of deformities and asymmetries that are sometimes undetected, the freedom to simulate distinct surgical procedures to obtain optimal results for the patient, and facility in evaluating and correcting the centric relation in the temporomandibular joint [52,53,54]. Although the costs are considerably higher, this approach might be considered a valuable option especially for adult patients suffering from severe OSAS.

The patient-centered outcomes in research should highlight QoL for emphasizing the need to understand health as a “state of physical, mental, and social well-being” [30,55]. Current literature suggests individuals with OSAS have poorer overall QoL compared to their healthy peers [30]. Further, it has been reported that poor QoL is related to the negative impact of OSAS on physical health outcomes and psychosocial functioning [6,7,8,9,10,11,12,13,14,15,16,17,18,19,20,30]. In this report, post-operative QoL evaluation was based on the questions taken from the OSAS specific domains [40,41,42,43]. These OSAS domains included post-operative evaluation of sleep quality (e.g., choking while sleeping, sleep disturbance, restless sleep); daytime function/activity (e.g., excessive drowsiness, poor attention span); emotional situation (e.g., emotional distress, mood swings, depression); physical symptoms (e.g., frequent colds; mouth breathing, tiredness); and work activity. According to the results of this report, MMA surgery appears to be associated with positive changes in OSAS-specific QoL domains (Out of 18 × 5 = 90 questions 76 answers pointed out “YES—Better” (85%), with the remaining 14 as “SAME—pre-op and post-op”, and no results as “NO—Worse”). Additionally, the patients declared that they were more satisfied with their facial appearance and their BMI decreased critically, which shows an improvement in physical and mental health status following orthognathic surgery.

## 5. Conclusions

According to the results of this study, maxillomandibular advancement surgery seems to be a safe and effective treatment option with beneficial results in terms of patients’ satisfaction and better quality of life in cases of severe OSAS in adult patients.

## Figures and Tables

**Figure 1 ijerph-19-06273-f001:**
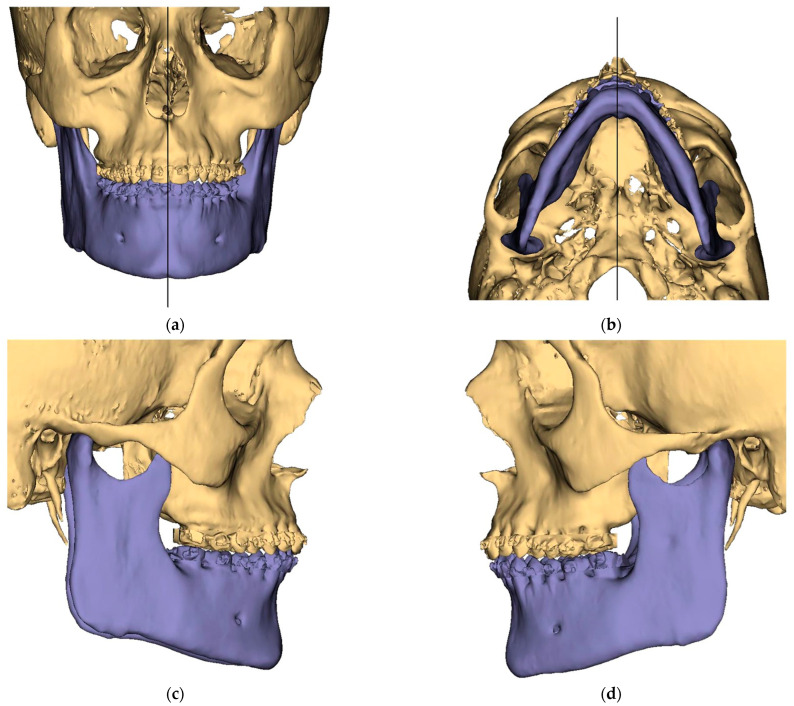
Surgical Plan: Preoperative situation: (**a**) Frontal view; (**b**) Occlusal view; (**c**) Right side view; (**d**) Left side view.

**Figure 2 ijerph-19-06273-f002:**
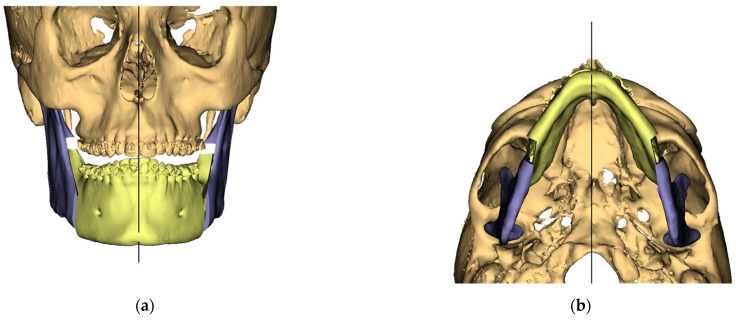
Surgical Plan: Intermediate position showing Mandibular Movement First. (**a**) Frontal view; (**b**) Occlusal view; (**c**) Right side view; (**d**) Left view.

**Figure 3 ijerph-19-06273-f003:**
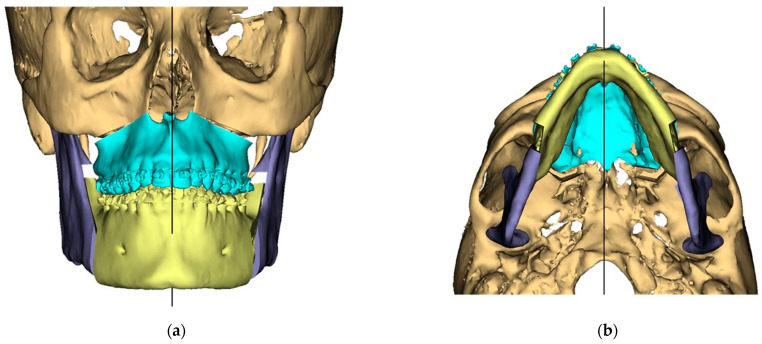
Surgical Plan: Final Position—Mandible moved according to planned maxilla position (Proximal segments rotated in) (**a**) Frontal view; (**b**) Occlusal view; (**c**) Right side view; (**d**) Left side view.

**Figure 4 ijerph-19-06273-f004:**
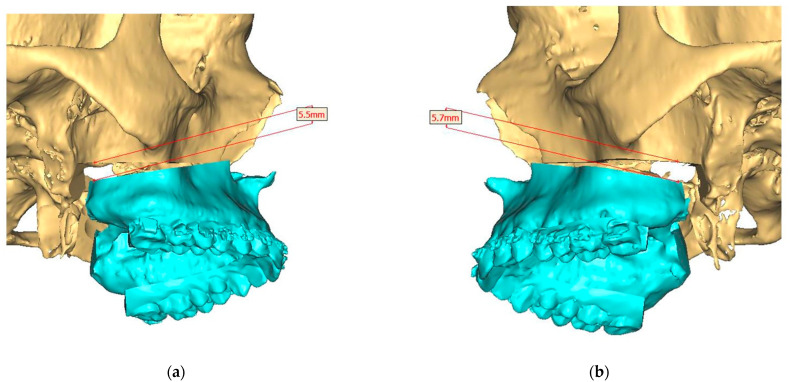
Surgical Plan: Maxilla movement overview (osteotomy thickness of 0.1 mm: (**a**) Right side view; (**b**) Left side view; (**c**) Frontal view.

**Figure 5 ijerph-19-06273-f005:**
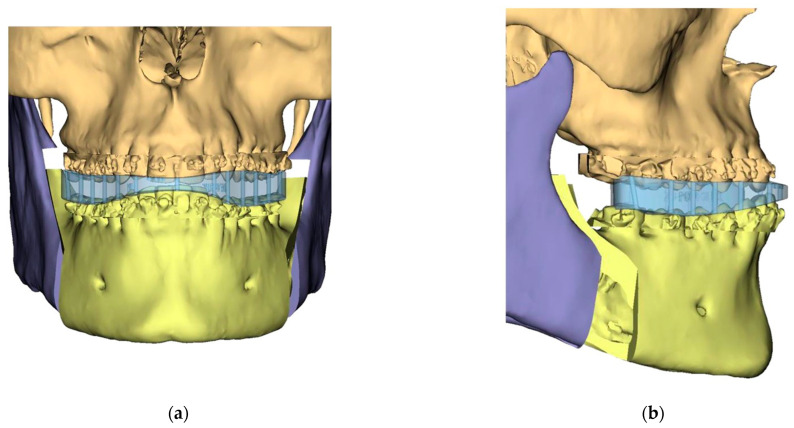
Guide Design: Intermediate Splint (Mandible surgery first)—Mandible has been auto-rotated slightly to allow for the design of the intermediate splint: (**a**) Frontal view; (**b**) Left side l view; (**c**) Splints.

**Figure 6 ijerph-19-06273-f006:**
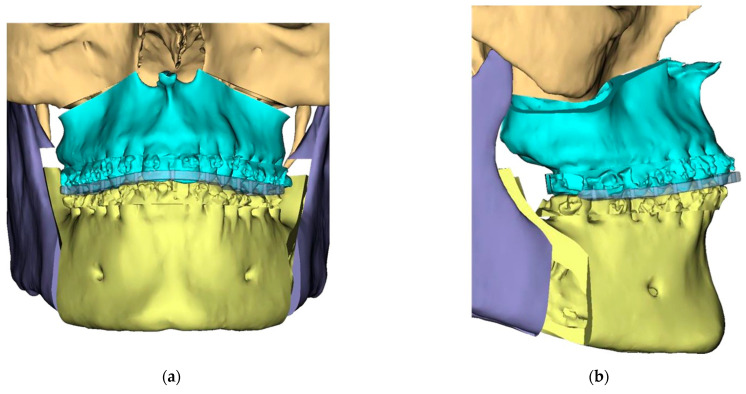
Guide Design and Final Splints: (**a**) Frontal view; (**b**) Right side view; (**c**) Splints.

**Figure 7 ijerph-19-06273-f007:**
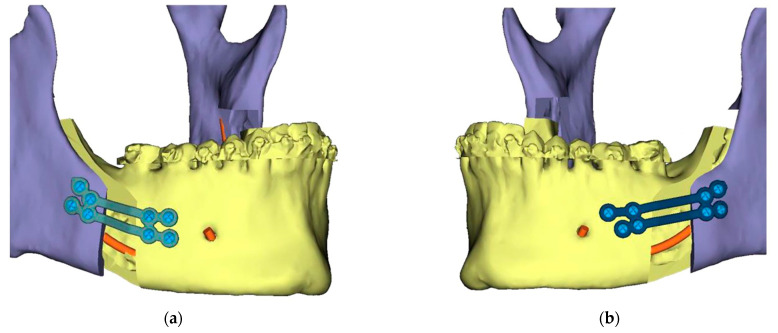
Titanium 3D Printed Plate for Mandible for use with Matrix ORTHOGNATHIC Ø1.85 mm screws—All screw pre-drilling guided using surgical guides. (**a**) Right side view; (**b**) Left side view.

**Figure 8 ijerph-19-06273-f008:**
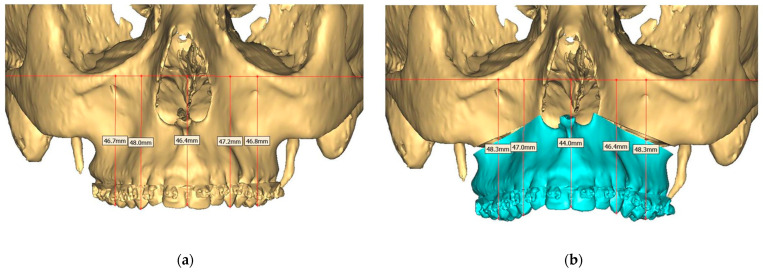
(**a**) Pre-operative and (**b**) Planned Maxilla.

**Figure 9 ijerph-19-06273-f009:**
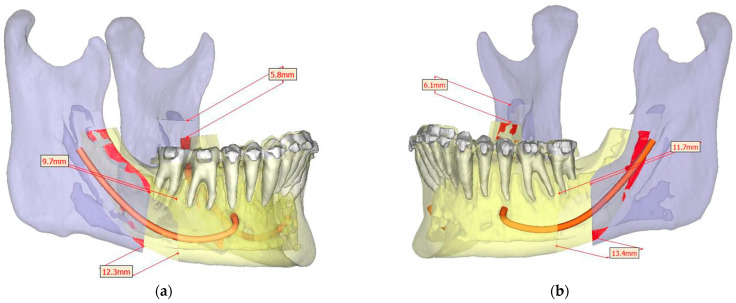
Surgical Plan: Mandible Movement Overview: (**a**) Right side view; (**b**) Left side view.

**Figure 10 ijerph-19-06273-f010:**
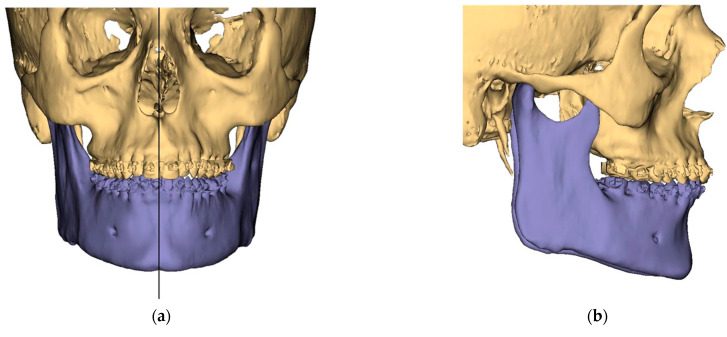
Surgical Plan: Before and After (**a**) Frontal view before; (**b**) Right side view before; (**c**) Frontal view after; (**d**) Right side view after.

**Table 1 ijerph-19-06273-t001:** Rustemeyer’s questionnaire about patients’ satisfaction after surgery.

Questions: Please Mark One Grade of the Scale from 0 (Poor) to 10 (Excellent)
R1. How would you assess your facial aesthetics before surgery?
R2. How would you assess your facial aesthetics after surgery?
R3. How would you assess your chewing function before surgery?
R4. How would you assess your chewing function today?
R5. How do you feel exactly about the surgical outcome of your operation?
R6. How do your relatives and friends feel in total about the surgical outcome of your operation?
0–1–2–3–4–5–6–7–8–9–10 (RESULTS AS 0 to 10)

**Table 2 ijerph-19-06273-t002:** Post-operative Quality of life Questionnaire.

Questions: Please Answer the Following Questions Based on Your Post-Operative Situation as: Yes Better/No Worse/Same
Q1. How would you assess your post-operative quality of sleep? Better than the pre-operative period?
Q2. How would you assess your post-operative day time function/activity? Better than the pre-operative period?
Q3. How would you assess your post-operative emotional situation? Better than the pre-operative period?
Q4. How would you assess your post-operative physical OSAS symptoms, if any such as breathing, frequent colds, tiredness etc). Better than the pre-operative period?
Q5. How would you assess your post-operative work activity? Better than the pre-operative period?

**Table 3 ijerph-19-06273-t003:** Patient Demographics.

Patient Number	Age at the Day of Surgery	Gender	General Health Condition (Other Than OSAS)	BMI INDEX (Pre-Op)	BMI INDEX (Post-Op)
1	30	M	Healthy	22.7	22.7
2	42	M	Healthy	37.6	35.9
3	59	M	Healthy	24.3	24.3
4	50	M	Healthy	35.4	31.9
5	58	F	Hypertension and tachycardiac	25.3	26.6
6	46	M	Hypertension (ACE inhibitor + calcium antagonist Perindopril + amlodipine-Takawita 8 mg) and Hyperhomocysteinemia (VitB + folic acid, Anti-aggregation drug—Cardirene 75 mg)	27.2	27.2
7	45	M	Healthy	27.8	30.9
8	33	M	Healthy	24.1	24.1
9	39	M	Healthy	30.7	30.7
10	53	M	Hyperuricemia (Allopurinol) andrenal colic	26.6	23.7
11	47	M	Hypertension	26.9	25.1
			CAD/CAM group mean value	28.05	27.55
12	39	M	Healthy	26.3	27.2
13	39	M	Healthy	25.2	27
14	44	M	Healthy	33.1	25.7
15	51	M	Hypertension and Diabetes	33.9	29.4
16	23	M	Healthy	34.6	28.1
17	38	M	Asthma (Relvar)	27.8	25.9
18	48	M	Hypertension (Pritor 20 mg)	33.6	31.3
			Traditional group mean value	30.64	27.80
			total	29.06	27.65

Patients 1–11: CAD/CAM group; Patients 12–18: Traditional group; M: Male; F: Female.

**Table 4 ijerph-19-06273-t004:** Characteristics of surgical interventions and list of complications.

Patient Number	Inter-Positional Bone Grafting in Maxilla	Genioplasty	Traditional Saw/Piezoelectric Surgery	CAD/CAM/Traditional Surgery	Post-Operative Complications
1	None	Yes	Stryker Saw	CAD/CAM	None
2	None	No	Stryker Saw	CAD/CAM	Open bite on right side
3	Iliac Crest	No	Stryker Saw	CAD/CAM	None
4	Iliac Crest	No	Stryker Saw	CAD/CAM	None
5	None	No	Stryker Saw	CAD/CAM	None
6	None	Yes	Stryker Saw	CAD/CAM	Hypoesthesia in lower lip (Vitamin B12 was prescribed and remission after 6 months)
7	None	No	Stryker Saw	CAD/CAM	None
8	None	No	Stryker Saw	CAD/CAM	None
9	None	No	Stryker Saw	CAD/CAM	None
10	None	No	Stryker Saw	CAD/CAM	None
11	None	No	Stryker Saw	CAD/CAM	Post-operative edema and ecchymosis. Bilateral permanent hypothesia in mandibular 3rd region.
12	None	No	Stryker Saw	Traditional	None
13	Iliac Crest	No	Stryker Saw	Traditional	None
14	Iliac Crest	No	Piezoelectric	Traditional	The mandible did not integrate after surgery. Revision surgery was scheduled and performed with successful results.
15	None	No	Stryker Saw	Traditional	TMJ problems including pain at mouth opening. Arthrocentesis and Botox injections at masseter muscle)
16	None	No	Piezoelectric	Traditional	None
17	Iliac Crest	No	Stryker Saw	Traditional	None
18	Iliac Crest	No	Stryker Saw	Traditional	Permanent hypoesthesia on superior lip right side and bilaterally in lower lip.

Patients 1–11: CAD/CAM group; Patients 12–18: Traditional group.

**Table 5 ijerph-19-06273-t005:** Rustemeyer’s questionnaire results.

Patient Number	R1	R2	R3	R4	R5	R6
1	6	8	7	8	9	8
2	10	10	8	10	10	10
3	7	7	9	7	8	7
4	6	8	8	4	7	9
5	5	7	6	5	8	8
6	6	9	7	5	10	9
7	10	10	10	10	10	10
8	7	8	8	8	10	10
9	7	9	6	10	9	10
10	7	9	8	5	10	9
11	9	7	9	9	9	9
12	7	8	6	9	10	10
13	6	6	8	5	7	7
14	8	6	9	5	10	10
15	8	10	8	9	10	10
16	7	7	7	7	8	8
17	7	7	6	8	8	9
18	6	5	7	7	9	8
Mann-Whitney test	1.00	0.06	0.40	0.78	0.74	0.89

Patients 1–11: CAD/CAM group; Patients 12–18: Traditional group.

**Table 6 ijerph-19-06273-t006:** QoL questionnaire results evaluating post-operative life quality.

Patient Number Yes (Better)/No (Worse)/Same (Post-Op vs. Pre-Op)	Q1	Q2	Q3	Q4	Q5
1	Yes better	Yes better	Yes better	Yes better	Yes better
2	Yes better	Yes better	Yes better	Yes better	Yes better
3	Yes better	Same	Same	Yes better	Same
4	Yes better	Yes better	Yes better	Yes better	Yes better
5	Yes better	Yes better	Yes better	Yes better	Yes better (less fatigue and less headache)
6	Yes better	Yes better	Yes better	Same	Yes better
7	Same (wife says better)	Same	Yes better	Yes better	Yes better
8	Yes better	Yes better	Yes better	Yes better	Yes better
9	Yes better	Yes better	Yes better	Yes better	Same
10	Yes better	Yes better	Yes better	Yes better	Yes better
11	Yes better	Yes better	Same	Yes better	Yes better
12	Yes better	Yes better	Yes better	Yes better	Yes better
13	Yes better	Same	Same	Yes better	Same
14	Yes better	Same	Yes better	Yes better	Yes better
15	Yes better	Yes better	Yes better	Yes better	Yes better
16	Yes better	Yes better	Same	Yes better	Yes better
17	Yes better	Yes better	Yes better	Yes better	Yes better
18	Yes better	Yes better	Yes better	Yes better	Yes better
Fisher’s exact test results	0.61	0.38	0.38	0.61	0.38

Patients 1–11: CAD/CAM group; Patients 12–18: Traditional group.

## Data Availability

Data of this work is available upon request.

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
