# Peer review of "Post-Operative Patients’ Satisfaction and Quality of Life Assessment in Adult Patients with Obstructive Sleep Apnea Syndrome (OSAS)"

_ijerph, 2022, doi:10.3390/ijerph19106273_

Round 1
Reviewer 1 Report
Dear Authors,
The article: 'Post-operative Patients` satisfaction and Quality of Life assessment in adult OSAS' was to evaluate the post-operative impact of bi-maxillary surgery on satisfaction and consequently the quality of life of these patients.
In my opinion, the title should expand the acronym OSAS.
English language and style must be inproved.
Numerous punctuation mistakes should be corrected.
Add email addresses for affiliates.
The introduction is well written. The purpose of the work should be clearly defined.
Materials and methods
Add information about an ethics committee (NUMBER!).
Figures should be with the signature. Correct Fig. 1.
Tables should be preparaed using MDPI guidelines.
The P value must be in italics.
The discussion is well written
line 84: 'Mthe OSA-18 questionnaire by Franco et al. is the most widely used QoL survey for paediatric OSAS and has been validated as an informative instrument [30, 40-43]' should be 'the OSA-18 questionnaire by Franco et al. [42] is the most widely used QoL survey for paediatric OSAS and has been validated as an informative instrument'. Check the citations in the text.
Discussion is clearly presented.
Add table with abbeviations befeore references.
Correct authors contributions using abbreviations of surnames.
Article can be accepted after major revision.
Author Response
REVIEWER 1:
The article: 'Post-operative Patients` satisfaction and Quality of Life assessment in adult OSAS' was to evaluate the post-operative impact of bi-maxillary surgery on satisfaction and consequently the quality of life of these patients.
In my opinion, the title should expand the acronym OSAS.
ANSWER: The title was adjusted as requested.
English language and style must be inproved.
ANSWER: English was rechecked and corrected for any mistakes.
Numerous punctuation mistakes should be corrected.
ANSWER: Punctuation was rechecked and corrected for mistakes.
Add email addresses for affiliates.
ANSWER: Corrections were applied
The introduction is well written. The purpose of the work should be clearly defined.
ANSWER: Thank you very much for your precious comment. In fact in the original text, we had two additional paragraphs which was accidentally erased during copy/pasting them to the template of the journal. Corrections were applied with additional references
Materials and methods
Add information about an ethics committee (NUMBER!).
ANSWER: The number was already written in acknowledgments. However, it was also added to M&M as requested.
Figures should be with the signature. Correct Fig. 1.
ANSWER: Figures were adjusted to fit the page with their legends. We were not able to understand the correction that is asked as signature to Fig 1.
Tables should be preparaed using MDPI guidelines.
ANSWER: Tables were adjusted
The P value must be in italics.
ANSWER: It was corrected
The discussion is well written
line 84: 'Mthe OSA-18 questionnaire by Franco et al. is the most widely used QoL survey for paediatric OSAS and has been validated as an informative instrument [30, 40-43]' should be 'the OSA-18 questionnaire by Franco et al. [42] is the most widely used QoL survey for paediatric OSAS and has been validated as an informative instrument'. Check the citations in the text.
ANSWER: Corrections were applied
Discussion is clearly presented.
Add table with abbeviations befeore references.
ANSWER: Abbreviations were added before references.
Correct authors contributions using abbreviations of surnames.
Article can be accepted after major revision.
Reviewer 2 Report
Dear Authors,
this paper is really interesting and well written.
Just few issues need to be addressed:
Abstract: abstract is really well written and understandable.
Introduction: introduction in well written, but it is really short. it should be improved, especially in lines 50 to 55 where references like this could help: Ludovichetti FS, Signoriello AG, Girotto L, Del Dot L, Piovan S, Mazzoleni S. Oro-dental lesions in paediatric patients with coeliac disease: an observational retrospective clinical study. Rev Esp Enferm Dig. 2022 Feb 16.
Materials and methods: well written, just one question: why 38 patients were selected? amu statistical issue?
Results: perfectly described
Discussion: well written, maybe the concept of patient's dissatisfaction and its origin could be addressed a little more.
Thank you
Author Response
REVIEWER 2:
Dear Authors,
this paper is really interesting and well written.
Just few issues need to be addressed:
Abstract: abstract is really well written and understandable.
Introduction: introduction in well written, but it is really short. it should be improved, especially in lines 50 to 55 where references like this could help: Ludovichetti FS, Signoriello AG, Girotto L, Del Dot L, Piovan S, Mazzoleni S. Oro-dental lesions in paediatric patients with coeliac disease: an observational retrospective clinical study. Rev Esp Enferm Dig. 2022 Feb 16.
ANSWER: Thank you for your contribution. Two paragraphs were added to the introduction, however we were not able to add the mentioned reference to the text because the topic of the mentioned reference was mainly on coeliac disease in child patients, while the aim of this work concentrated on adult OSAS patients.
Materials and methods: well written, just one question: why 38 patients were selected? amu statistical issue?
ANSWER: 38 patients were chosen according to the inclusion criteria, mainly as adult OSAS patients that are operated in our clinic. We didn’t do sample size calculation for such a retrospective study, they were all patients treated consecutively in the period indicated in the paper.
Results: perfectly described
Discussion: well written, maybe the concept of patient's dissatisfaction and its origin could be addressed a little more.
ANSWER: As can be seen on Table 5, the patients have given numbers from 5 to 10 as an evaluation for their satisfaction. All their answers were above 5, so in our opinion, these answers cannot be considered as dissatisfaction. Additional explanation was added to the text.
Reviewer 3 Report
This is a perfectly written manuscript with beautiful illustrations and exhaustive references, for which one can only congratulate. Unfortunately, the title needs to be lengthened considerably, as hardly any physicians know what OSAS is:
"Postop..... in Patients with Obstructive Sleep Apnea Syndrome (OSAS) after Maxillo-Mandibular Advancement."
Line 99: delete the word "were" and put "Le" on the next page
Line117: reduce the total size of the figures a bit so that the legend is on the same page
Line 362-364 delete the whole paragraph Acknowledgment.
Thanks for letting me read it !
Author Response
REVIEWER 3:
This is a perfectly written manuscript with beautiful illustrations and exhaustive references, for which one can only congratulate. Unfortunately, the title needs to be lengthened considerably, as hardly any physicians know what OSAS is:
"Postop..... in Patients with Obstructive Sleep Apnea Syndrome (OSAS) after Maxillo-Mandibular
ANSWER: Thank you for your suggestions that can improve our work.
The tittle was corrected as asked =
Advancement."
Line 99: delete the word "were" and put "Le" on the next page
Line117: reduce the total size of the figures a bit so that the legend is on the same page
Line 362-364 delete the whole paragraph Acknowledgment.
ANSWER: These corrections were applied as requested
Round 2
Reviewer 1 Report
line 300 skip to next page